# Relationships between Coping Styles, Emotional Distress, and Fear of COVID-19 among Workers in the Oil and Gas Industry in Malaysia during the COVID-19 Pandemic

Joseph Kulip [1], Mohammad Saffree Jeffree [1,*], Nicholas Tze Ping Pang [1,*], Nazmirrudin Nasirruddin [1] and Walton Wider [2]

1   Faculty of Medicine and Health Sciences, Universiti Malaysia Sabah, Kota Kinabalu 88400, Sabah, Malaysia; jokulip89@yahoo.com (J.K.); bm18160097@student.ums.edu.my (N.N.)
2   Faculty of Business and Communication, INTI International University, Nilai 71800, Negeri Sembilan, Malaysia; walton.wider@newinti.edu.my
*   Correspondence: saffree@ums.edu.my (M.S.J.); nicholas@ums.edu.my (N.T.P.P.)

**Abstract:** The COVID-19 pandemic has had serious impacts on psychological health globally. However, very little is currently known regarding the link between fear of COVID-19 with psychological health and various coping styles, especially among oil and gas workers. This study aims to assess the prevalence of depression and anxiety among oil and gas workers, and subsequently examine the role of sociodemographic and occupational variables, various coping styles, and emotional distress in contributing to fear of COVID-19. A total of 299 oil and gas workers participated in this study. The DASS-21, Brief COPE, and Fear of COVID-19 (FCV-19) were used to assess the research variables. The descriptive analyses of DASS-21 indicated a prevalence of 26.8%, 33.5%, and 17.1% for depression, anxiety, and stress, respectively, among oil and gas workers. The results also indicated that all types of coping styles (problem-oriented, emotion-oriented, and dysfunctional-oriented) were significant predictors of fear of COVID-19. Sociodemographic and occupational variables and emotional distress variables were not significant predictors of fear of COVID-19. The study suggests how crucial it is for occupational mental health surveillance and prompt intervention for oil and gas workers.

**Keywords:** mental health; COVID-19 fear; oil and gas industry; coping styles; depression; anxiety

## 1. Introduction

The COVID-19 pandemic has created multiple psychological issues especially with isolation and loneliness due to multiple lockdowns and quarantines [1]. One of the unseen epidemics has been that of loneliness and potential depression in oil and gas workers. Oil and gas workers, by the nature of their jobs, have to spend long periods on isolated oil rigs with no contact with their families. This was amplified exponentially during the pandemic, as lockdowns meant oil and gas workers were either marooned for indefinite periods on their rigs or were only allowed to return to hotels or quarters on shore. There was no recourse to going to their home states due to strict inter-state testing and quarantine protocols. This alone would be a significant factor in potentially resulting in increased psychopathology; depression and anxiety would thus be a major occupational health problem, as there will be lower levels of productivity, higher levels of absenteeism and the converse phenomenon of presenteeism [2], and potential risks among workers operating difficult and complex machinery due to concentration and attention difficulties.

The previous literature in oil and gas workers' pre-pandemic is also scant, with minimal good quality epidemiological data available. One of the seminal studies had 1472 subjects comprising a mix of offshore and onshore workers. In offshore workers, the prevalence of potential anxiety was 10.2% and 1.2% for anxiety, whereas for onshore workers the prevalence of potential anxiety and anxiety was 11.55% and 2.4% [3]. Depression

statistics were higher, with offshore workers having 14.7% potential depression and 2% depression, and onshore workers showing 17.7% potential and 5.1% actual depression. Corresponding data from the Middle Eastern oil and gas industry suggest roughly a 15% prevalence rate of both anxiety and depression [4], whereas recent study from an Indonesian setting within the COVID-19 pandemic demonstrated paradoxically far lower levels between 2.4 and 2.5% for anxiety and depression [5].

However, there has been no evidence among the oil and gas worker population that looks further than merely prevalence, but that also looks at how non-psychological and psychological variables can be correlated to fear of COVID-19. Fear of COVID-19 is a new construct in the literature unique to COVID-19 that encompasses both components of depression and anxiety, as well as phobias [6]. Sociodemographic variables, coping styles, and emotional distress have been identified in the literature as risk factors for fear of COVID-19 [7–10]. Specifically, studies suggest that higher education levels are associated with higher fear of COVID-19 [11]; whereas females [12] and older age groups [13] tend to experience greater fear of COVID-19. More updates regarding knowledge and information about COVID-19 might make one more fearful during pandemic [11]. Additionally, numerous coping strategies have been identified to reduce concerns about being infected with COVID-19. Task- or emotion-oriented coping have been identified as effective in reducing fear of COVID-19 [10]. The literature also suggests that psychological factors such as emotional distress are associated with fear of COVID-19 but in a more specific university population [6]. Research has yet to uncover whether there are relationships between emotional distress and fear of COVID-19 in the context of oil and gas workers.

Hence, this study aims to not only ascertain the prevalence in a Malaysian oil and gas worker population, which will allow us to get a better estimate of both psychopathologies in oil and gas worker populations, but also to explore the underlying psychological process that drives the fear of COVID-19 in a pandemic-stricken population. Specifically, we identify the prevalence of depression and anxiety, using survey data collected from oil and gas workers in Sabah state, Malaysia. Next, we examine the relationship between sociodemographic and occupational variables, coping styles, depression, anxiety, stress, and fear of COVID-19.

## 2. Materials and Methods

### 2.1. Participants

This was a cross-sectional study which was performed over a time period of two months from 1 December 2021 to 31 January 2022. An online survey link was shared across various platforms. As traditional convenience sampling method is known to be less generalizable and precise than a homogeneous convenience sampling method, which can prompt estimation bias [14], we therefore adopted a homogeneous sampling strategy in the current study. Oil and gas workers are a homogeneous group that have different occupational characteristics compared to other occupations.

A total of 299 participants were enrolled in the study, of which 98.3% were male ($n$ = 294). Of the sample, 76.3% were married ($n$ = 228), and 60.5% had completed at least secondary education ($n$ = 181). In addition, 94.2% were offshore workers ($n$ = 282), while the number of permanent and contract staff were evenly split. In terms of working shift, the majority (46.8%) worked more than 28 days ($n$ = 140). Descriptive analyses of DASS-21 suggested that 73.2% of the participants had a normal level of depression ($n$ = 219); 66.5% had a normal level of anxiety ($n$ = 199); and 82.9% had a normal level of stress ($n$ = 248) (see Table 1)

**Table 1.** Descriptive analysis of participants (N = 299).

|  |  | Mean | Frequency | Percent |
|---|---|---|---|---|
| Age |  | 37.3 years old |  |  |
| Gender | Male |  | 294 | 98.3 |
|  | Female |  | 5 | 1.7 |
| Marital Status | Married |  | 228 | 76.3 |
|  | Single |  | 63 | 21.1 |
|  | Single Parent |  | 8 | 2.7 |
| Educational level | Secondary |  | 181 | 60.5 |
|  | Tertiary |  | 118 | 39.5 |
| Job operation | Offshore |  | 282 | 94.3 |
|  | Onshore |  | 17 | 5.7 |
| Job status | Permanent |  | 147 | 49.2 |
|  | Contract |  | 147 | 49.2 |
|  | Part-time |  | 5 | 1.7 |
| Working shift | 14 days |  | 59 | 19.7 |
|  | 28 days |  | 100 | 33.4 |
|  | More than 28 days |  | 140 | 46.8 |
| Depression | Normal |  | 219 | 73.2 |
|  | Mild |  | 45 | 15.1 |
|  | Moderate |  | 6 | 2.0 |
|  | Severe |  | 18 | 6.0 |
|  | Extra Severe |  | 11 | 3.7 |
| Anxiety | Normal |  | 199 | 66.5 |
|  | Mild |  | 1 | 5.7 |
|  | Moderate |  | 44 | 14.7 |
|  | Severe |  | 4 | 1.3 |
|  | Extra Severe |  | 29 | 9.7 |
| Stress | Normal |  | 248 | 82.9 |
|  | Mild |  | 16 | 5.4 |
|  | Moderate |  | 10 | 3.3 |
|  | Severe |  | 18 | 6.0 |
|  | Extra Severe |  | 7 | 2.3 |

### 2.2. Procedure

Most participants were recruited at the official health screening station prior to going offshore at the airport, hence an almost universal sampling of all consenting oil and gas workers going offshore between that period was performed. The inclusion criteria were workers in the oil and gas industry either offshore or onshore who were based at Sabah state, Malaysia, which is one of the largest sources of offshore oil production in Malaysia. The exclusion criteria included workers who are under psychiatric follow up or having acute medical problems that precluded them from answering the questionnaires, or workers who had just started employment during the period of questionnaire distribution.

### 2.3. Ethical Considerations

Ethical consideration and approval to conduct the study was obtained from the Institutional Review Board of Universiti Malaysia Sabah [(protocol code 2/21 (10)].

### 2.4. Instruments

Four questionnaires were employed in this study. Firstly, a sociodemographic (e.g., age, gender, marital status, educational level) and occupational (e.g., job status, working shift, job operation) questionnaire was employed to collect data for descriptive analysis.

### 2.4.1. The Depression, Anxiety, and Stress Scale-21 Items (DASS-21)

The Depression Anxiety, and Stress Scale 21 (DASS-21) Malay version was used to assess data on emotional distress [15]. The DASS-21 consists of 21 items and three dimensions: depression (e.g., "I felt that I had nothing to look forward to"); anxiety (e.g., "I felt I was close to panic"); and stress (e.g., "I found it difficult to relax"). For each item, respondents were requested to answer using a four-point scale, ranging from 0 (did not apply at all) to 3 (applied a lot or most of the times). The Cronbach alpha coefficient ranged from 0.96 to 0.97.

### 2.4.2. Fear of COVID-19 Scale (FCV-19S)

Next, the Fear of COVID-19 scale (FCV-19S) Malay version was employed to ascertain levels of fear of COVID-19 [16,17]. The FCV-19S has seven items (e.g., "I am most afraid of coronavirus-19"). Respondents were requested to answer using a five-point Likert scale ranging from 1 (strongly disagree) to 5 (strongly agree). A greater score represents extreme fears of COVID-19. The Cronbach alpha coefficient was 0.95.

### 2.4.3. Coping Orientation to Problems Experienced (Brief-COPE) Inventory

A Malay version of the Brief-COPE [18] was used to assess workers' coping styles with pandemic consequences. The Brief-COPE consists of 28 items and three sub-scales: problem-oriented coping (e.g., "I've been thinking hard about what steps to take"); emotion-oriented coping (e.g., "I've been expressing my negative feelings"); and dysfunctional coping (e.g., "I've been giving up trying to deal with it"). For each item, respondents were requested to answer using a four-point scale, ranging from 1 ("I have not been doing this at all") to 4 ("I have been doing this a lot"). Cronbach's alpha coefficient ranged from 0.93 to 0.95.

### 2.5. Data Analysis

Data analysis was performed using SPSS Version 27. Descriptive statistics were described using mean, frequency, and percentage for age, gender, marital status, educational level, job operation, job status, working shift, depression, anxiety, and stress level among respondents. We inspected skewness ($\pm 3$) and kurtosis ($\pm 10$) indices for the normality assumption [19]. Subsequently, Pearson's r correlation coefficients were calculated between all study variables at a bivariate level. Multicollinearity was inspected by assessing tolerance and variance inflation factor (VIF). Linear multiple regression analysis was performed with fear of COVID-19 as the dependent variable, with the level of significance set as $p < 0.05$. R-squared changes were reported for the proportion of variance in fear of COVID-19. In this study, gender, marital status, educational level, job operation, and job status were set as dummy variables to ensure the accuracy.

## 3. Results

A Pearson's r correlation coefficients analysis was performed to examine for significant relationships between sociodemographic variables, occupational variables, coping styles, depression, anxiety, stress, and fear of COVID-19. Based on Table 2, the Pearson's r correlation coefficient results suggested that fear of COVID-19 is significantly correlated with educational level (secondary), depression, anxiety, stress, problem-oriented, emotion-oriented coping, and dysfunctional coping. Specifically, the strongest association with fear of COVID-19 came from dysfunctional coping ($r = 0.539$, $p < 0.01$), followed by problem-oriented coping ($r = 0.477$, $p < 0.01$), emotion-oriented coping ($r = 0.401$, $p < 0.01$), stress ($r = 0.373$, $p < 0.01$), depression ($r = 0.363$, $p < 0.01$), anxiety ($r = 0.352$, $p < 0.01$), and educational level (secondary). Due to high correlations between depression, anxiety, and stress, we combined the three dimensions and operationalised it as emotional distress. Hence, a multiple linear regression was performed with fear of COVID-19 as the dependent variable; whereas educational level (secondary), emotional distress, problem-oriented coping, emotion-oriented coping, and dysfunctional coping were employed as the predictors.

Before the multiple regression analysis was conducted, the attention was focused to ensure the four key assumptions of multiple regression, namely homoscedasticity, linearity, multicollinearity, and normality were tested. Based on Table 2, the values of skewness and kurtosis were in the acceptable ranges [19]. However, gender, marital status, and job operation had non-normal distributions owing to a few outliers of more male, married, and offshore workers in our samples. The tolerance was above 0.1 and the variance inflation factor (VIF) was below 10, therefore, multicollinearity was not an issue in our research variables [20]. Hence, multiple regression analysis was performed once all the assumptions were achieved [21].

**Table 2.** Pearson correlation results.

| No | Variables | 1 | 2 | 3 | 4 | 5 | 6 | 7 | 8 | 9 | 10 | 11 | 12 | 13 |
|----|-----------|---|---|---|---|---|---|---|---|---|----|----|----|----|
| 1 | Depression | 1 | | | | | | | | | | | | |
| 2 | Anxiety | 0.949 ** | 1 | | | | | | | | | | | |
| 3 | Stress | 0.950 ** | 0.938 ** | 1 | | | | | | | | | | |
| 4 | Gender (Female) | −0.010 | 0.007 | −0.037 | 1 | | | | | | | | | |
| 5 | Age | −0.034 | −0.087 | −0.019 | 0.101 | 1 | | | | | | | | |
| 6 | Marital status (Married) | −0.021 | −0.021 | −0.055 | −0.088 | −0.363 ** | 1 | | | | | | | |
| 7 | Educational level (Secondary) | −0.044 | −0.026 | −0.003 | −0.162 ** | 0.074 | −0.085 | 1 | | | | | | |
| 8 | Job status (Permanent) | −0.103 | −0.086 | −0.126 * | 0.031 | −0.251 ** | 0.145 * | −0.128 * | 1 | | | | | |
| 9 | Job operation (Off-shore) | −0.006 | −0.026 | 0.010 | −0.306 ** | 0.025 | 0.044 | 0.275 ** | −0.079 | 1 | | | | |
| 10 | Problem-oriented coping | 0.458 ** | 0.494 ** | 0.496 ** | −0.136 * | −0.045 | −0.007 | 0.021 | −0.071 | 0.113 | 1 | | | |
| 11 | Emotion-oriented coping | 0.418 ** | 0.454 ** | 0.464 ** | −0.142 * | 0.035 | 0.002 | 0.077 | −0.093 | 0.108 | 0.924 ** | 1 | | |
| 12 | Dysfunctional coping | 0.627 ** | 0.637 ** | 0.626 ** | −0.044 | −0.074 | 0.029 | −0.128 * | −0.084 | 0.009 | 0.830 ** | 0.782 ** | 1 | |
| 13 | Fear of COVID-19 | 0.363 ** | 0.352 ** | 0.373 ** | −0.012 | 0.061 | −0.079 | −0.138 * | −0.056 | 0.065 | 0.477 ** | 0.401 ** | 0.539 ** | 1 |
| | Skewness | 1.706 | 1.634 | 1.419 | −7.576 | 0.286 | 2.661 | 0.433 | 1.062 | 3.847 | 0.720 | 0.476 | 1.372 | 0.656 |
| | Kurtosis | 2.454 | 2.334 | 1.488 | 55.765 | −0.396 | 8.303 | −1.825 | 2.616 | 12.89 | −0.365 | −0.870 | 1.711 | −0.429 |

Note: ** $p < 0.01$; * $p < 0.05$.

Linear multiple regression analysis was performed to identify the variables that explained fear of COVID-19. The results showed an $\Delta R^2$ value of 0.302 for fear of COVID-19 was acceptable, which accounted for 30.2% of the exploratory variance. Based on Table 3, dysfunctional coping ($\beta = 0.405$, $p = < 0.000$) was the greatest significant predictor for fear of COVID-19. This was followed by problem-oriented coping ($\beta = 0.382$, $p = 0.008$) and emotion-oriented coping ($\beta = -0.286$, $p = 0.028$) as additional predictors of fear of COVID-19. Educational level (secondary) and emotional distress were not significant predictors. The results of the multiple linear regression model analysis are summarized in Table 3.

**Table 3.** Predictors of fear of COVID-19 among oil and gas workers.

| Dependent Variable | Predictors | β | t | p | ΔR² | F | p | TOL | VIF |
|---|---|---|---|---|---|---|---|---|---|
| Fear of COVID-19 | Educational level | −0.070 | −1.370 | 0.172 | 0.302 | 26.81 | 0.000 | 0.891 | 1.122 |
| | Problem-oriented coping | 0.382 | 2.683 | 0.008 | | | | 0.116 | 8.636 |
| | Emotion-oriented coping | −0.286 | −2.208 | 0.028 | | | | 0.140 | 7.157 |
| | Dysfunctional coping | 0.405 | 3.899 | 0.000 | | | | 0.217 | 4.604 |
| | Emotional distress | 0.050 | 0.789 | 0.431 | | | | 0.575 | 1.739 |

## 4. Discussion and Conclusions

These results suggest a prevalence of depression and anxiety (using a cut-off point of mild) of around 27% and 33.5%, respectively. Necho et al. (2021) in their meta-analysis using a general population reported a higher prevalence of depression and anxiety with 34.3% and 38.1%, respectively [22]. Therefore, it can be seen that the level of awareness of oil and gas workers in the present study is considered higher than the general community with regard to COVID-19-related information. However, our findings reported a higher prevalence of depression and anxiety among oil and gas workers compared to other similar studies. In Indonesia, ref. [5] reported the prevalence of depression and anxiety of oil and gas workers at 2.4% and 2.5%, respectively. Fitriana et al.'s (2022) intra-pandemic study in Indonesia was performed in the early months of the pandemic (end of 2020), hence the psychological effects of depression and anxiety may not have been evident so quickly [5]. The present study was performed almost one and a half years following the beginning of COVID-19 lockdowns in Malaysia. Hence, the cumulative effect of multiple lockdowns, loss of economic opportunity, long periods of isolation offshore and onshore, and the actual fear of the illness of COVID-19 itself would have had opportunity to manifest itself as depression or anxiety [23,24]. As there are no other contemporaneous studies other than the Indonesian study examining depression and anxiety in a time of COVID-19, this would be a pioneer study in documenting the psychological consequences of COVID-19 in a very specific but crucial population. There are multiple evidence-based interventions, adapted from the Malaysian setting, which can be delivered as brief 10–15 min interventions that can be done by non-professional healthcare workers [25].

Interestingly, the variance in fear of COVID-19 is mostly contributed to by all three coping styles, and not by emotional distress. This suggests that there is utility in teaching oil and gas workers simple acts of self-care in order to promote the use of emotion-oriented coping styles, and to avoid the use of dysfunctional coping styles [26]. The use of dysfunctional methods to avoid the pain of living in a pandemic such as substance and alcohol use, distraction techniques, and avoidance techniques do not take away the fear of COVID-19 in the long term; they merely cause it to disappear momentarily [27]. Thus, basic problem-solving skills can be taught as a crucial primary prevention method for mental health [28] in order to decrease the fear of COVID-19. Emotional distress only contributed 0.5% to the variance of fear of COVID-19 and was not significant. This suggests that psychological process variables, rather than psychopathology or emotional distress per se, would contribute more significantly to the overall fear of COVID-19, and can be an important adjunctive quality to examine in future moderation or mediation studies.

Although various limitations may preclude this study from achieving its highest level of statistical power, directions for future research are acknowledged. Firstly, there is scant

literature out there related to oil and gas workers and the psychological sequelae. There is also only one extant paper detailing the psychological consequences of COVID-19 in oil and gas workers. However, it used a different set of psychological instruments, which would hence have different case-finding ability. Thus, this research project would be useful in serving as a benchmark for the actual prevalence of both depression and anxiety in a specific oil and gas population, and hence further research to re-ascertain prevalence once COVID-19 has passed to an endemic phase would be essential to calculate "peacetime" estimates of prevalence. Secondly, as this study only examined individuals who were working in Sabah, which has slightly different ethnic makeups compared to the rest of Malaysia, the prevalence of depression and anxiety in different populations may vary. Thirdly, as the fear of COVID-19 scale is a newly developed instrument, it does not yet have a categorical cut-off point of any clinical utility, which can be the focus of future research.

Despite the limitations stipulated above, the current findings propose practical implications by providing the association between sociodemographic, occupational variables, various coping styles, emotional distress, and fear of COVID-19. In view of the negative impacts of COVID-19 on oil and gas workers' psychological mental health, exploring coping styles is a crucial step to developing intervention strategies to deal with fear of the COVID-19 pandemic. Hence, coping styles-based interventions might be useful to assist oil and gas workers in managing fear and in fostering their psychological mental health amidst COVID-19. Mental health providers should therefore integrate various coping styles in their psychological intervention among the oil and gas workers. For example, they can be taught to focus on problem-solving skills rather than using dysfunctional-type coping, such as alcohol consumptions, denial, and substance use as their coping mechanism. Furthermore, emotion-oriented coping strategies should also be promoted as a buffer against fear of COVID-19. The role of acceptance, emotional support, humor, religion, and self-blame should be adopted in the psychological intervention.

In conclusion, this study underscores the importance of various coping styles on oil and gas workers' fear of COVID-19. In light of this, factors such as problem-oriented coping, emotion-oriented coping, and dysfunctional coping are crucial. The dysfunctional coping in particular was found to have the largest influence. Although our study reported that emotional distress is not a predictive factor of fear of COVID-19, this finding needs to be cautiously interpreted as the psychological distress stemming from COVID-19 is persistent and pervasive [29], and does not disappear as lockdown and worsening wave hits Malaysia [30]. The fear of COVID-19 is a new construct that has only begun to be explored [31], therefore more research is warranted to identify the relationship of emotional distress and other factors with fear of COVID-19. Nevertheless, the high prevalence of depression and anxiety in the current study showed that most oil and gas workers are uniquely susceptible to mental health difficulties during COVID-19. Additionally, the fear of personal infection or infection of friends and family has proven to be an obstacle in occupational functioning. Thus, it is vital for mental health providers to confront these problems in order to be successful in providing psychological interventions to oil and gas workers and ultimately reducing fear of COVID-19.

**Author Contributions:** Conceptualization, J.K. and M.S.J.; methodology, N.T.P.P.; software, J.K.; formal analysis, N.T.P.P.; investigation, N.T.P.P.; resources, J.K.; data curation, J.K.; writing—original draft preparation, J.K. and N.T.P.P.; writing—review and editing, W.W.; visualization, N.N., W.W.; supervision, M.S.J.; project administration, J.K.; funding acquisition, M.S.J. All authors have read and agreed to the published version of the manuscript.

**Funding:** This research received no external funding.

**Institutional Review Board Statement:** The study was conducted in accordance with the Declaration of Helsinki, and approved by the Institutional Review Board of Universiti Malaysia Sabah (protocol code 2/21 (10), date of approval 17 August 2021).

**Informed Consent Statement:** Informed consent was obtained from all patients involved in the study.

**Data Availability Statement:** Data can be made available upon reasonable request.

**Conflicts of Interest:** The authors declare no conflict of interest. The funders had no role in the design of the study; in the collection, analyses, or interpretation of data; in the writing of the manuscript, or in the decision to publish the results.

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
