# Peer review of "Relationships between Coping Styles, Emotional Distress, and Fear of COVID-19 among Workers in the Oil and Gas Industry in Malaysia during the COVID-19 Pandemic"

_sustainability, doi:10.3390/su14095398_

Round 1

Reviewer 1 Report

The present manuscript analysis the relationships between Coping Styles, Depression, Anxiety, and Fear of COVID-19 Among Workers at Oil and Gas Industry  in Malaysia During COVID-19 Pandemic.

  • Moderate English changes required
  • Abstract: The reviewer suggests to write again the abstract especially highlighting the literature background psychological consequences of COVID-19. Furthermore, reviewer suggest inserting a short paragraph regarding the features discovered during the analysis within the 200 words limits.
  • Introduction: The reviewer suggests adding, at the end of the Introduction, a short paragraph related to how the manuscript is structured.
  • Materials and Methods: the reviewer suggests extending this section in the following ways
    • Questionnaires: they could be describe in more details
    • Data analysis: explain it in more details especially regarding the variables used and the method adopted
  • Results: they should be better explained in much more details and they should be better formatted
  • Sections Discussion and Conclusions should be merged into one and also the reviewer suggests adding a paragraph about further implications and research lines derived from the present work

Reviewer 2 Report

Thank you for asking to review “Relationships between Coping Styles, Depression, Anxiety, and Fear of

COVID-19 Among Workers at Oil and Gas Industry in Malaysia During COVID-19 Pandemic”.  This was an interesting topic. However, I do have some concerns to be addressed.

Method

- Is there any randomized sampling method used to reduce the selection bias?

- It may be better to mention if any or no need the institutional review board approval.  

Results

-Table 1

 It will be clear to list the important variables like mean age, male proportion, major diseases which are risk factors related to depressive disorders and anxiety disorders.

Discussion

  • Comparing the prevalence of depression and anxiety with general population may provide more information about these workers.

To conclude, this is an interesting study and may be a potentially important contribution to both existing knowledge and practical aspects of the field. I recommend accepting the paper.

Reviewer 3 Report

  1. See line 88-89: “majority of the participants showed normal level of depression, anxiety, and stress with 73.24%; 66.5%; and 82.9% respectively”. This kind of expression may produce an ambiguity . “73.24%; 66.5%; and 82.9%” are not “normal level”, they are constituent ratios.
  2. This research only focus on “Four questionnaires” and no “providing basic psychological interventions” (line 177-178). By logic, the conclusions (“ this study underscores the importance of case detection and mental health screening in oil and gas workers, who are uniquely susceptible to mental health difficulties due to their isolation and loneliness. Research should be focused on providing basic psychological interventions to oil and gas workers, which are founded on solid evidence, and are able to be delivered via telehealth to obviate the need to have dedicated mental health personnel on board each oil rig.”, line175-180) cannot be drawn from the results in the article.
  3. Failure to draw appropriate conclusions based on results.

Reviewer 4 Report

the paper is sound and safe. The conclusions are in agreement with the proposed research model. If it is possible (if data are available of course) some considerations concerning the impact of two supplementary dimension (SarsCov 2 variants infected people rates among oil workers and or the rates of infection on their families) may further increase the readers interest

Round 2

Reviewer 1 Report

  •  Moderate English changes required
  • Introduction: the reviewer suggests improving this part of the manuscript with more information about the the topic of the research
  • Material and methods: the reviewer suggests write again iin a more extensive way this section
  • Results: the reviewer suggests write again iin a more extensive way this section 

Author Response

  • Moderate English changes required
    • We have revised the English language.
  • Introduction: the reviewer suggests improving this part of the manuscript with more information about the the topic of the research
    • Additional information have been added in the introduction section marked with red font.
  • Material and methods: the reviewer suggests write again iin a more extensive way this section
    • Data analysis section has been revised extensively. 
  • Results: the reviewer suggests write again iin a more extensive way this section 
    • Results has been revised extensively. 

Reviewer 3 Report

No more

Author Response

We are grateful for your consideration of this manuscript, and we also very much appreciate your suggestions, which have been very helpful in improving the manuscript.

Round 3

Reviewer 1 Report

  1. Sub-section 4.2 should be included within the main text without specify the beginning of this subsection
  2. Conclusions: should be included some further research implementations
  3. Section 3: the reviewer suggests delating the sub-headings and consequently write again this section
  4. Section 2 : improve the sub-section Participants

Author Response

1. Sub-section 4.2 should be included within the main text without specify the beginning of this subsection 

- Thank you examiner. The sub-section 4.2 has been removed.

2. Conclusions: should be included some further research implementations

- Thank you examiner. We have included further research implementations in the conclusion part.

3. Section 3: the reviewer suggests delating the sub-headings and consequently write again this section

- Thank you examiner. The sub-headings in section 3 has been removed and the section has been re-write.

4. Section 2 : improve the sub-section Participants

- Thank you examiner. The 4.1 Participant has been improved.